

# GC Insights: Lessons from participatory water quality research in the upper Santa River basin, Peru

Sally Rangecroft[1,2], Caroline Clason[2,3], Rosa Maria Dextre[4], Isabel Richter[5], Claire Kelly[2], Cecilia Turin[6], Claudia V. Grados-Bueno[4], Beatriz Fuentealba[4], Mirtha Camacho Hernandez[4], Sergio Morera Julca[7],
John Martin[2] and John Adam Guy[2]

[1] School of Geography, Faculty of Environment, Society and Economy, University of Exeter, Exeter, UK
[2] School of Geography, Earth and Environmental Sciences, University of Plymouth, Plymouth, UK
[3] Department of Geography, Durham University, UK
[4] Instituto Nacional de Investigación en Glaciares y Ecosistemas de Montaña (INAIGEM), Huaraz, Peru
[5] Department of Psychology, Norwegian University of Science and Technology, Trondheim, Norway
[6] Instituto de Montaña, Lima, Peru
[7] Instituto Geofísico del Perú (IGP), Lima, Peru

*Correspondence to*: Sally Rangecroft (s.rangecroft@exeter.ac.uk)


**Abstract.** Research around water security in the Peruvian Andes rarely includes a local perspective or engages in a participatory approach with local communities within the research process. Here we share four key lessons from an interdisciplinary project that gathered community perspectives on water quality issues in the upper Rio Santa basin. Mixed-methods data was collected via a photo elicitation app with a survey (Nuestro Rio), and a field work campaign. Our main
learnings were i) the importance of in-person engagement; ii) the accessibility of technology for data collection; iii) the need for co-produced knowledge and solutions; and iv) the complexity of water quality as an environmental concept. Our research highlights the importance of effective participant engagement methods to support socio-environmental integration to support sustainable decision-making and water resource management.

## 1 Introduction

Water quality is a key consideration for basic survival as well as both socio-economic and environmental sustainability. However, due to both natural events and human activities, water quality is threatened in many regions of the world, also intensified by impacts of climate and land use change (Anderson, 2016; Magnússon et al., 2020). Water quality directly impacts the lives of water users (Azevêdo et al., 2022), yet local perspectives, knowledge, and emotions are often not considered (Dextre et al., 2022). Furthermore, water *quality* can be a secondary consideration to water *quantity*, despite being closely
intertwined in water insecurity (Clason et al., 2023; Rangecroft et al., 2023). Water quality is a complex, multifaceted issue, which can be judged by indicators such as acidity, clarity, smell, taste, or chemical composition (Flotemersch and Aho, 2021).



Some aspects of water quality are visible (e.g. colour, turbidity or litter), whereas others are "hidden" (e.g. heavy metal content) and typically only detectable by field or laboratory instrumentation and analysis (Flotemersch and Aho, 2021). Whilst water quality variables are often measured and monitored through methodologies commonly available in the natural sciences, local

communities can provide unique information about the state of their ecosystem (Okumah et al., 2020, Richter et al., 2022). Local water users not only directly depend on local water sources, but are also sensitive to changes in water availability, quality, and ecosystems over time, contributing to traditional ecological knowledge (TEK), and providing insights beyond the temporal and spatial scope of in-situ measurements (Pauly, 1995; Azevêdo et al., 2022). Furthermore, community participation and TEK can help decision makers to develop feasible solutions and facilitate tailor-made governance that is accepted and

implemented by multiple parties (Mistry and Berardi, 2016; Albagli and Iwama 2022, Richter et al., 2022).

Here we present four key lessons from the Nuestro Rio project, an interdisciplinary and international collaboration that explored local community perspectives on water quality in the glaciated upper basin of the Santa River in Peru. The Santa River is incredibly important for food-water-energy security, both locally and regionally (Baraer et al., 2012; Recharte et al.,

2017), experiencing both water quality and availability issues related to glacier retreat and anthropogenic pressures (e.g. pollution; extraction; water governance) (Magnússon et al., 2020; Aylas-Quispe et al., 2021).

## 2 Methods

To gather insights on water quality from local communities, we applied a multi-method approach comprising of both an app for uploading pictures and responding to survey questions, and face-to-face interviews. The Nuestro Rio app was itself

designed as a mixed methods approach to gather insights into local perceptions of water quality, in addition to identification of perceived water quality issues and their drivers in this region (for more details about the survey see Appendix A). Geolocated photographs of local waters and associated survey data from participants were collected through the app. The app was designed in Spanish as the most accessible language for the study region. To facilitate data collection, researchers directly communicated with several communities across the upper Santa River basin (Fig. 1). Participants (aged 18+) were invited to engage with the

app on tablets provided by the researchers (e.g. ownership of smart devices and advanced technological skills were not required), offering guidance, and translation where needed. Semi-structured interviews were also conducted but are not considered in this article (for more information see Rangecroft et al. (2023)). Quantitative data was analysed using descriptive statistics, and qualitative data coded and analysed using an emergent thematic framework to identify key themes. Throughout this research and community engagement process, key lessons were learnt which will be shared here as the focus of this insight

paper.





**Figure 1: Simplified map of perceived water (collected via the Nuestro Rio app) quality across the study area of the upper Santa River basin, Peru. Points outlined by colour to represent the participant-rated water quality (good, neutral, bad) with example photos from participants.**






## 3. Key lessons

### 3.1 The importance of in-person engagement

Engaging directly with local participants during fieldwork proved immensely valuable, outweighing financial and time expenditures. This approach also mitigated potential research fatigue in the region, by emphasising the quality of interactions over their quantity. The significant majority of the 350 data entries collected were the result of direct community engagement in the field. Participant engagement is known to be a challenge for citizen science and participatory data collection (Fraisl et al., 2022), as reflected in the poor app uptake by those not involved with field activities. Direct in-person interactions can address potential obstacles, whether they relate to limited access to, or familiarity with, smart devices (as discussed in section 3.2) or issues of trust. In-person fieldwork also allowed for informal dialogues with participants, providing clarity on the survey questions or research aims if needed (see section 3.4).

### 3.2 Challenges of digital (in)accessibility

Considering potential barriers such as device reception, technological familiarity, lack of access to smartphones, and perceptual challenges is critical when introducing an app for data collection in a diverse community setting. There are numerous reasons why individuals might not have utilised the Nuestro Rio app on their personal devices during our study. Originally, we planned in-person training sessions for the app, but due to COVID-19, we had to transition to training videos and online workshops, which may have limited the potential for recruitment and promotion outside of community-specific fieldwork. This suggests that researchers should consider the target audience's comfort with installing and using technologies such as apps, and ideally co-develop apps with participant groups (Daum et al., 2019). Deliberate training strategies are also recommended (e.g. online, on-site, handout instructions) (Martin et al., 2021; Fraisl et al., 2022). A considerable portion of the local population either lack smartphones or are not tech-savvy, an issue intensified by factors of demographics and intersectionality. The language used by the app (Spanish) could have been an additional barrier for participants in rural areas where Quechua is the primary language for many people. There was also a lack of internet access in rural areas, restricting digital accessibility even for those more familiar with smart devices. Conversely, while urban regions had better connectivity, the app's adoption rate remained low among younger users, potentially due to insufficient engagement incentives. Future research could further disentangle motivations for participation in regions where technology is a limitation, as well as regions where it is not a limitation.

### 3.3 Need for co-produced knowledge and solutions

Sustainable approaches to water management typically require an engaged community (Dean et al., 2016). However, a critical first step to building this engaged community is to understand how the community perceives their water resources and the management thereof, an issue rarely examined in research (Steinwender et al., 2008; Dean et al., 2016; Okumah et al., 2020). Our work indicated a community desire for engagement and openness to co-design of solutions, in addition to a desire to communicate their perceptions to local and regional decision makers. Local communities are often not included in decision





making processes around management of water resources in this region (Dextre et al., 2022), yet actively engaging with communities can be an entry point to inclusive resource management. Furthermore, engaging local people in the decision-making process itself can help to empower individuals and communities to influence water governance processes and also

strengthen the acceptance and support for new resource management policies (Okumah et al., 2020; Albagli and Iwama, 2022). The Nuestro Rio project was well-received by participants who were familiar with past research initiatives in the region in which they had often felt neglected. The app provided a vehicle for participants to make their perspectives known and contribute to knowledge generation. In designing and implementing a project, it is vital to think beyond publication of results and one-way dissemination, and to consider continuous, inclusive research efforts. From our work together in designing and

sharing results, we have learned how important it is to include local experts at each step for true global collaboration. It is essential to make sure everyone has an equal say, especially in projects between the North and South where power asymmetries can be present. Furthermore, it is vital that we return the knowledge generated to the participants who helped build it.

### 3.4 Complexity of "water quality" as an environmental concept

The term water quality was interpreted differently by groups of participants, and often required clarification during field-based

data collection, especially in rural areas. This discrepancy might stem from ambiguous communication of research objectives, participants' more pressing concerns, or varying terminologies concerning water across different languages and cultures. For instance, some languages might have specific terms for unique types of water or aspects of water quality that lack direct translations. This nuanced understanding underscores the importance of face-to-face interactions and the potential for qualitative data methods, such as interviews, to bridge the comprehension gap between participants and researchers. A

Quechuan-Spanish translator assisted during field-based data collection to support use of preferred language, however translation was not possible for participants engaging with the app independently. Language and cosmovision are thus other key considerations, given indigenous perceptions of and emotions related to water, in addition to the importance of water beyond its role as a physical resource (Tipa, 2009; Azevêdo et al., 2022).

### 4. Recommendations for future participatory environmental research

The lessons learned from this project offer important considerations for design of future community engagement for co-production of knowledge and solutions around environmental issues. A shift away from heavy reliance on monitoring and modelling data and towards a more integrated approach, covering insights from both the natural and the social sciences, for environmental assessment is required for equitable and sustainable resource management (Drenkhan et al., 2023). Effective design of participatory research can play a crucial role here. Collecting local perceptions through digital technology potentially

allows for wider uptake, however experiences from the Nuestro Rio project demonstrate that in-person engagement was essential for maximising participant response. Adapting approaches to different settings requires a deep understanding of the local context, emphasising the importance of familiarising oneself with the local landscape from the initial research design

phase. In many contexts, digital-only approaches cannot encapsulate the deeper understanding obtained through participant-researcher dialogue. It is essential to thoroughly consider and address logistical challenges in data collection. This includes

ensuring accessibility in terms of location, technology, and language; embedding research and researchers within communities; catering to community needs; and grasping the driving factors, or lack thereof, behind participation. Effective communication along the community-research-policy-management continuum requires careful consideration of how data are understood and valued. Additionally, it is vital to acknowledge the varied timeframes across research, decision-making, and execution phases. Researchers should recognise the temporal scale required for developing an understanding of place, stakeholder relationships,

and building of community trust (Rangecroft et al., 2021). Finally, there is also an important distinction to be considered here between citizen science and participatory research for giving participants *agency* in influencing decision-making (Albagli and Iwama 2022; Illingworth, 2023).

**Author contribution:**

All authors are the Nuestro Rio research project team, and enabled the research to be possible through various contributions.

The manuscript was developed from insights discussed in project meetings and conversations related to other project outputs. SR & CC led on manuscript preparation with editing contributions from IR, CK and RDM. RMD prepared the manuscript figure.

**Competing interests:**

The authors declare that they have no conflict of interest.

**Ethical statement:**

The research was conducted with ethical approval from the University of Plymouth. Considerations of good ethical practice included gaining informed consent for participation, only including participants aged 18 or over, and the anonymity of data. Other areas of good ethical practice included the dissemination of results and outputs back to involved communities and participants where possible.


**Acknowledgements:**

The authors would like to thank all the participants for their time and interest, as without them the project would not be possible. The Nuestro Rio project was funded by GCRF via the University of Plymouth, and further supported by the Newton Fund (UK NERC grant number NE/S013245/1) and ProCiencia-CONCYTEC (Peru contract number 010-2019-Fondecyt). The authors

would also like to extend their thanks to all those who helped to support in the field, their involvement and support was crucial.



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



**Appendix A**: **Nuestro Rio survey**

*[Please note that this is the English translation for Appendix 1; the original questions on the Nuestro Rio app were in Spanish]*

START

[Participant to take photo of water]


SECTION 1 – About your photograph

Q1: What is this river or stream called? (Do you have any other names for it?)  [open text]

Q2. On a scale of (1) "very bad" to (5) "very good", how would you rate the water quality of the river or stream?
1 Very bad
2 Bad
3 OK (Neutral)
4 Good
5 Very good
Don't know

Q3: Why do you think the water quality of the river or stream is like this? There is no right or wrong answer, we just want to hear your opinion. If you are not sure, please just write "I don't know".  [open text]


SECTION 2 – Your photograph

What do you feel when you see the water of the river or stream? For each of the following moods, please record how you feel on a scale of (1) Not at all, to (5) Extremely.


Q4. ANGRY?
1. Not at all
2. A little
3. Neither yes, or no (Neutral)
4. Very



5. Extremely

Don't know

Q5. AFRAID?

1. Not at all

2. A little

3. Neither yes, or no (Neutral)

4. Very

5. Extremely

Don't know

Q6. HAPPY?

1. Not at all

2. A little

3. Neither yes, or no (Neutral)

4. Very

5. Extremely

Don't know

Q7. SAD?

1. Not at all

2. A little

3. Neither yes, or no (Neutral)

4. Very

5. Extremely

Don't know

Q8. SURPRISED?

1. Not at all

2. A little

3. Neither yes, or no (Neutral)

4. Very

5. Extremely

Don't know




SECTION 3 – About you

Q9. Where do you live (name of place)?  [open text]


Q10. What is your age?  [open text]

Q11. Please indicate your gender:  [multiple choice]

Male

Female

Choose not to identify

Other

Q12. What is your MAIN occupation – CHOOSE ONE [multiple choice]

Agriculture/livestock

Teaching

Student

Commerce/business

Public administration (Civil servant)

Mining (or related)

Transport

Household work

Other

Other please specify (Optional)  [open text]

END