# Peer review of "GC Insights: Lessons from participatory water quality research in the upper Santa River basin, Peru"

_EGUsphere, 2023_

## Author Response (AR1)

**Authors response**

*We thank the reviewer for their extensive and constructive comments. We have revised our manuscript accordingly. Please find below our point-by-point responses to the reviewers' specific comments in blue italic throughout with reference to specific changes in our manuscript where appropriate.*

**Reviewer 1**

**General comments**

This research aims to explore participatory water resource research approaches in the Peruvian Andes at the interface of science, community and decision-making. As stated by the authors, these approaches have barely been implemented in common – often natural-science-dominated – water resource research. Such an endeavour is highly complex and requires careful research design and local participation since early project implementation. While overall, I find this study well-written, a topic highly relevant and timely, and important for future research agendas in the Peruvian Andes and beyond, I have several concerns regarding the depth and meaningfulness of the presented findings and "key lessons" mentioned by the authors. Therefore, I suggest a thorough revision of the described methods, their scope and related outcomes in order to more strongly work out most relevant and novel key findings according to the experiences and data collected in the field.

*We thank the reviewer for their comments. We would like to emphasize that we have submitted this article as a short Insights piece, rather than a full research article. We have already published a full research article exploring the data and findings in detail (Rangecroft et al., 2023) and here we opted for a much shorter piece to look to communicate insights from the app development and implementation itself, which can be extremely important for researchers considering developing technological tools for encouraging participation and engagement. We want to make sure this is clear as we believe it shows that a number of these reviewers suggestions are not possible/relevant given the focus, limited word count and limited reference list possible for a Geoscience Communications Insights article. For example, due to the nature of this short article (1,500 words), we were not able to provide a full description of the wider and deeper process of research tool development (including piloting the tool with local students) that went into the creation of the Nuestro Rio app.*

*From reflecting on all three reviewers' comments, we have reframed our piece, which we believe has addresses a number of suggestions from reviewer 1 in their recommendation to bring the "most relevant and novel key findings according to the experiences and data collection in the field". In this context, our revised manuscript focuses on the development and implementation of a novel technological tool (e.g. the Nuestro Rio app) for facilitating a move towards participatory science.*

*We would also like to emphasize that the audience for Geoscience Communications is extremely broad, and whilst certain key lessons might be well known for some disciplines (e.g. the importance of in-person engagement, the need for co-produced knowledge and solutions) this is not necessarily the case across the board. We believe that the lessons included in our Insights piece can help those who are considering developing and utilizing novel technology for research, particularly those who may not have engaged with participatory environmental research previously.*

*We have now reframed this article to really emphasize the focus on the learnings of the technological approach. This can be exampled with the new abstract:*

*"Here we share four key lessons from an interdisciplinary project (Nuestro Rio) that gathered community perspectives on local water quality in the Rio Santa basin (Peru) utilising a digital technological approach where we collected data via a novel photo elicitation app, supported with a field work campaign. The lessons explored in this article provide insights into challenges and opportunities for researchers considering developing technological tools for encouraging participation and engagement in marginalised communities."*

**Specific comments**

Abstract:

From a first read, some of the presented "main learnings" (or "key lessons" as stated afterwards) do not seem striking to me to justify the meaningfulness of this research. The "importance of in-person engagement" (to include local stakeholders) is broadly known – but difficult to achieve – and the "need for co-produced knowledge and solutions" is in most cases the key pillar of participatory research and decision-making processes. Highlighting these aspects as two out of four key findings this research has produced, probably misses the opportunity to provide local insights – whether specific to the catchment or valid beyond – and discuss major bottlenecks and perspectives to advance participatory research in a meaningful way as desired by many but probably only achieved by a few researchers in this field.

*Due to the limited focus of this Geoscience Communication Insights article, we refrained from providing more local insights as we wanted to make the key learnings from development and implementation of the Nuestro Rio app as transferable and applicable to other settings as possible. We explore our specific research findings, and what these mean in a local context, in much more detail in our Rangecroft et al., 2023 paper, which we will have now signposted to more clearly in our revised article (e.g. lines 44, 59-60).*

Methods:

The multi-method approach with the Nuestro Rio app seems quite novel and interesting to me. However, I have several doubts and concerns, probably some of them owed to the short description of them.

*We thank you for your comments here, but we would like to emphasize that this was submitted as a Geoscience Communications Insights article, with a short word limit of 1,500 words, so detailed discussion of our methods was not the intended focus of the article. However, we would like to highlight that the methods applied in our research are described in detail within our Rangecroft et al., 2023 paper, and that is now more clearly signposted to in our revised article (e.g. lines 44, 59-60).*

1) It is not clear if the research design has been part of a co-production process with local stakeholders since the beginning of the project, or if the entire research framework including the app and interviews were developed by the researchers only. If I understand right, you have only used quantitative-qualitative data and descriptive statistics (lines 57-58) for this development. If

this is true, then the lack of key stakeholder participation in the research design stage should be discussed as a possible shortcome or bias. Or in other words: relevant questions not being asked cannot (necessarily) be identified afterwards. This also represents a point for future improvement (section 4).

*We outline the project and methods in much more detail in our Rangecroft et al., 2023 paper. However, for context - the research here was designed with a range of in-country partners who are already deeply engaged with the case study communities, and the research approach and data collection tools were developed with those partners and were based on their extensive knowledge of the issues already identified by the case study communities in their prior work. We have tried to include this in our revised manuscript, such as lines 82 – 85: "This change highlights the importance of designing technology with the target audience's needs in mind, potentially through co-developing apps with participants (Daum et al., 2019). Such a participatory process however requires extended research project periods, particularly for complex, transdisciplinary projects 85 such as Nuestro Rio."*

2) It is not clear why the semi-structured interviews – in most qualitative research a key pillar to produce local insights – have been conducted but not considered in this paper. What are the reasons behind? Would the findings from the interviews not have substantially contributed to construct more in-depth key findings as discussed above?

*Thank you for the interest in this qualitative data, but due to the very limited word count and focus of this Insights paper, we can only signpost to our JoH paper (Rangecroft et al., 2023) which explores the data and results from the interviews in much more detail, allowing us to keep this paper focussed on the use of technology for facilitating participatory research.*

3) Considering the experiences of (several Peruvian!) authors of this research with indigenous communities, why have you not opted for creating an app version in Quechua language? As stated by the authors (lines 111-116), language is a barrier and might considerably influence the quality of responses, willingness to participate and thus lead to biased results, particularly in regions and communities where Quechua is the mother language and Spanish still perceived as the colonial or upper-urban class language. You cannot change this at this point anymore but should clearly attribute this fact as a potential shortcome and bias to your results, and recommend such approaches to future research.

*We opted to develop the app in Spanish rather than Quechua to try and reach a wider population in the basin, since many residents of Huaraz do speak Spanish. This decision was made by the research team, with informed opinion from those with experience working in the region. We would also like to verify that our in-field researchers were able to support the translation of the text in the Nuestro Rio app into Quechua by conducting walking interviews with participants using the app. More can be read on this in Rangecroft et al. 2023.*

*We have tried to ensure that this is more clearly stated in our revised version, for example in line 53: "The app was designed in Spanish as the most accessible language for the study region." And the further discussion on potential limitations related to language in lines 74-76: "Although the app's use of Spanish potentially posed a barrier in areas where Quechua is the dominant language, our in-field researchers facilitated communication by translating materials to and from Quechua (Rangecroft et al., 2023)."*

4) Please include a brief statement about the selection and representativeness of the participants. What were the criteria to approach them? Were you able to include a broad intersection of

community members in terms of age, gender, socio-economic status, local water resource challenges, among others?

*As for our responses regarding methodology discussed above, we would like to reiterate that within this 1,500 word article we did not have the space to explore participant recruitment, and details about our participants can be found in Rangecroft et al., 2023.*

5) Was the "return of knowledge" to the communities (lines 107, 148-149), actually an important aspect for long-term work, research ethics and communications, part of your project design? Which activities were implemented to guarantee a maximal return flow of knowledge, implications and lessons learnt, and thus benefit to local communities?

*This was a very short pilot project, and whilst the return of knowledge to the communities was conducted via return visits and a travelling photo exhibition, the project itself did not have capacity to build upon the initial round of participatory data and research. Due to the short nature of this 1,500 word article, we have not explored this further here as our focus is the lessons learnt from the development and implementation of the Nuestro Rio app. In our revised version we have ensured that the project is clearly described as a pilot project (line 48) and we have included the statement (lines 102-104): "Furthermore, in project design and delivery it is vital that we return the knowledge generated to the participants who helped build it, however this can be extremely challenging in projects with limited funded time".*

6) The Figure and concrete links to the represented data in the text should be improved. Which patterns of water quality can be identified according the Nuestro Rio survey? There is not even one single reference to Figure 1 in your text. Does the variable size of the photos mean anything (i.e. better/worse water quality)? The photos are hard to read and a legend with symbols for each photo could help to understand which water quality issue has been identified in each case (if a sort of major "water quality issue categories" is feasible). Also add at least main cartographic elements, such as a scale and coordinates, at least for the small overview maps. It seems Figure 1 belongs to "Results" (or here: "Key lessons") rather than "Methods".

*Again, we would like to emphasize that the purpose of this short article was not to explore the findings themselves, and this information can be read in detail within Rangecroft et al., 2023. We have improved the figure caption to communicate that the figure is simplified to illustrate the project concept and not represent our results. The revised figure caption (lines 63 – 66) is: "Figure 1: Map of perceived water quality (collected via the Nuestro Rio app) across the study area of the upper Santa River basin, Peru. Point colour represents the participant-rated water quality (good, neutral, bad), and examples of photos from participants are shown. Note that this figure is simplified to illustrate the project concept, and is not a representation of our full, in-depth research findings."*

Key lessons:

As mentioned above (Abstract), I would strongly suggest to lift the key findings to a higher level. I would delete or rewrite sections 3.1 ("The importance of in-person engagement") and 3.4 ("Need for co-produced knowledge and solutions") which naturally represent the essence of participation, mutual exchange and knowledge production. Instead, you can further explain e.g. a) what the best strategies were to engage people (e.g. incentives you generally mention; to guarantee an environment where they are willing and feel comfortable to transparently respond), b) which limitations you were dealing with during the research (e.g. distrust and previous

negative experiences of communities with researchers; comprehension of the technical details and scientific language), and c) what the implications and tangible results of your research are (e.g. if the results can now directly be used for other research; what the return value of this research for local implications and benefit is).

Are most of the outcomes specific to the respective subcatchment conditions (e.g. local conflict over water resources; acid rock drainage) or can (some of them) be understood as key lessons for the Santa river basin and beyond?

*We have made edits throughout our revised manuscript in sections 3.1 The importance of in-person engagement and 3.3 Need for co-produced knowledge and solutions. Whilst we value all of these comments, it is outside the scope (and length) of this paper to reflect further on specific research results here, but we hope that the manuscript helps to encourage further discussion and thoughts on these aspects.*

Recommendations for future participatory environmental research:

I like several points raised here. One important point: how can the challenges from completely different timeframes (lines 133-135) of short research frameworks and publication pressure compared to long-term work with communities to understand each other and to build trust be overcome or at least improved?

Try to be as specific as possible within this short GC piece. Sentences such as "Effective design of participatory research can play a crucial role here" are quite general. They need to be put into clear context (e.g. what "effective" means; "can play" sounds quite vague).

As mentioned above, include the importance of co-produced research design and local language focus (Quechua) for future participatory science-community approaches.

*Thank you for these reflections. We have tried to be more specific with our language in our revised version as recommended. We believe that our edits have also helped to more clearly state the importance of local language and co-production, as described in responses above.*

**Technical corrections**

The text is well and technically correctly written.

One single comment: avoid colloquial or exaggerated language, such as "incredibly important" (line 44) or "immensely valuable" (line 68).

*Thank you for this correction, we have revised our language in these two situations in our revised version.*

**Reviewer 2**

**General Comments:**

This is a good start on an article intended to share lessons from a "participatory" project that elicited perceptions of water quality from several communities in the upper portion of the Rio Santa watershed in the Andes of Peru. With that said, it may be better to publish this manuscript as a research note than a scientific article and to change the framing of participatory to one of citizen science (quite different). While the subject matter of the manuscript is within the scope of Geoscience Communication, the manuscript does not place the research project and results within existing work on the several topics with which the authors engage. For example, the project is labeled as participatory, but it is unclear how residents of the Rio Santa valley - the Callejon de Huaylas - participated other than using an app given to them (on cellphones? not clearly stated in the methods section). Also, there is a limited engagement with the vast water governance literature for the Peruvian Andes. Furthermore, while this seems to be a citizen science project, there is no outlined citizen science framework for the reader to place this project in. It almost seems like a research project that did a limited literature review before engaging in "participatory" research ... On the other hand the conclusion, that points towards more integrated water quality monitoring which includes both data collection with instruments and citizen engagement to elicit perceptions of quality, is timely and important. This conclusion may be valid but needs to be better substantiated by placing the research in existing literature frameworks and providing more details on the "participatory" method used. For this reason I recommend a major revision.

Note: for all the suggested citations below, see who cites them in recent literature ...

*We thank you for your time, insights and comments. We would like to emphasize that we have submitted this article as a short Insights piece, rather than a full research article. We have already published a full research article exploring the data and findings in detail (Rangecroft et al., 2023) and here we opted for a much shorter piece to look to communicate insights from the app development and implementation itself. We want to make sure this is clear as we believe it shows that a number of these suggestions are not possible/relevant given the focus, limited word count and limited reference list possible for a Geoscience Communications Insights article. For example, due to the nature of this short article (1,500 words), we were not able to provide a full description of the wider and deeper process of research tool development (including piloting the tool with local students) that went into the creation of the Nuestro Rio app.*

*We would like to emphasize that participatory research can be and mean a range of things, but importantly here we don't claim any 'deep' participation. Instead, we highlight the need for more participatory methods to enable local voices to be heard, and we offer an example of how we have tried to do that in a small way using a novel app. In this article we specifically consider the important lessons - the advantages and the shortcomings - of developing and using technology for this purpose.*

*We have now reframed our article to ensure this is front and center, and to ensure that readers are aware of our full research article (Rangecroft et al., 2023) where they can learn about the project methods and findings in much more detail. For example, we have now signposted readers to Rangecroft et al. (2023) more clearly in our revised article (e.g. lines 44, 59-60).*

**Specific Comments:**

*Figure 1:*

The map is clear an has good design, but it is not clear why some photographs are bigger than others. It is also not clear what criteria was used to choose which points have photos. I found myself wanting a section in the methods to briefly describe how the survey instrument was used/analyzed to categorized water quality as good, neutral, or bad. Also how the sites were chosen, or how participants were chosen?

*Again, we would like to emphasize that the purpose of this short article was not to explore the findings themselves, and this information can be read in Rangecroft et al., 2023. We have now improved the figure caption to communicate that the figure is simplified to illustrate the project concept and not represent our results. The revised figure caption (lines 63 – 66) is: "Figure 1: Map of perceived water quality (collected via the Nuestro Rio app) across the study area of the upper Santa River basin, Peru. Point colour represents the participant-rated water quality (good, neutral, bad), and examples of photos from participants are shown. Note that this figure is simplified to illustrate the project concept, and is not a representation of our full, in-depth research findings."*

*Please also note that it was participants who rated the water quality, through their perception, not scientific instruments.*

*Lessons Learned:*

*In person engagement:* I would like to know how long the project lasted and what plans there are to make the engagement sustainable. For example, will there be an ongoing campaign to encourage people to continue to use the app? Was there "participation" in the design of the survey questions and the app itself? Was there "participation" in designing the project activities? Whatever the answer may be, it would be great to know a little more about this.

*This was a pilot project which lasted just over 1 year (2021-2022). The short time-frame is related to funding constraints, which means there is not an ongoing campaign. We have now made this explicitly clear in the revised version on line 48: "During a short pilot project (2021-2022)" and we discuss funding issues and the challenges of short project timeframes for inter- and transdisciplinary projects and co-design and co-production (e.g. lines 83-85, line 127).*

*The reframing of our manuscript means that we have focused on communicating our insights to help others who might be designing similar science communication and participatory research. For more details on the project, methods and findings, please see our published article Rangecroft et al., 2023.*

*Challenges of digital (in)accessibility:* I am fairly certain there is a decade long history of using smart-phones for various research project in this region (as an example, Bury et al. 2013. New geographies of water and climate change in Peru: Coupled natural and social transformations in the Santa River Watershed, Annals of the Association of American Geographers, 103(2), 363-374.) It would be great to link some of the previous work in this area to this water monitoring project. In the same way, there is plenty of literature on app design and problems with engaging citizens in digital data collection on devices (especially for map based apps, a good starting point might be Haklay, M., & Tobón, C. (2003). Usability evaluation and PPGIS: towards a user-centered design approach. International Journal of Geographical Information Science, 17(6), 577-592. https://doi.org/10.1080/1365881031000114107).

*We had very limited scope with the number of references we could draw upon for this short article (25 references maximum), which explains why we weren't able to fully represent the range of previous research that exists in this field. We would like to point out that the Nuestro Rio pilot project was not specifically about monitoring water quality, but instead about exploring local perspectives of water quality, and about exploring how a novel digital tool could be used for this purpose.*

*Need for co-produced knowledge and solutions:* I once again find myself wanting to see more engagement with existing literature on this topic. Maybe Ostrom, E. (1996). Crossing the great divide: Coproduction, synergy, and development. World Development, 24(6), 1073-1087. http://www.sciencedirect.com/science/article/B6VC6-3VW1PS3-9/2/ef9e7e3fd45a5cfda5b6b045ca20105e and/or Budds, J., & Hinojosa, L. (2012). Restructuring and Rescaling Water Governance in Mining Contexts: The Co-Production of Waterscapes in Peru. Water Alternatives, 5(1), 119-137. is a good place to start. I also really want to know how the perception data was combined with any existing measured water quality data (pH, turbidity, salinity, heavy metals, etc. etc. etc.). As noted in the conclusion, this combination may be a more powerful way to assess water quality ... co-production ...

*Please see our full research article, Rangecroft et al. 2023, for more information on our project findings. We would also like to re-emphasize that this was a pilot project that did not have the scope to combine social and physical data, although we agree that these could be a powerful way to assess water quality in future work.*

*Complexity of "water quality" as an environmental concept*: Yes, this is difficult. I have also worked with Andean communities and water quality. I struggle to see this project as participatory other than engaging informants (like old school ethnology). For many readers participatory might mean an interpreting reality together, the line between "participants" and "researchers" becomes blurred, but in this manuscript a division between the participants and researchers is maintained ... good starting points here might be Pain, R., & Francis, P. (2003). Reflections on participatory research. Area, 35(1), 46-54. https://doi.org/10.1111/1475-4762.00109 and/or Freire, P. (1970). Pedagogy of the Oppressed. Continuum. I find myself wanting to know what observations might have been made in the semi-structured interviews ... perhaps in these results are the insights for co-produced knowledge about water quality.

*Again, we would direct you to our published work which introduces the project, the methods and the findings in detail (Rangecroft et al., 2023), but we have now more clearly signposted to this in our revised version. In this GC Insights article our intention is to explore the lessons from our process of developing and implementing the Nuestro Rio tool for others who may be considering developing technological tools for encouraging participation and engagement with research. Participatory research is both a range of methods and an ideological perspective. It is fundamental that the subjects of the research become involved as partners in the process of the enquiry, and that their knowledge and capabilities are respected and valued. In this piece we are not claiming deep or substantial research co-design. We have offered an opportunity for local stakeholders to tell us, through imagery and words, about their understandings of their local water environments (participants were involved in choosing what, where and when to show us), and how they feel about these environments (we valued and respected their perspectives). We are aware that much of the previous research in this river basin has a focus on physical observations, and has often excluded local perspectives and experiences. It was our intention to explore the*

*value of these perspectives as indicators of water quality in their own right, providing a voice for social perspectives and knowledge within water quality research. We believe that the edits we have made to our revised version throughout have made this clearer.*

**Technical corrections:**

None, the paper is readable.

*Thank you.*

This might also be useful:

Goodchild, Michael F. (2007). Citizens as sensors: the world of volunteered geography. GeoJournal, 69(4), 211-221. doi: 10.1007/s10708-007-9111-y.

*Thank you for this suggestion. We were limited with the number of references we could include due to the nature of the manuscript being a Geoscience Communications Insights piece. We have included Saleem et al. (2024) in our revised version.*

**Reviewer 3**

**General Comments:**

The aim of this Geoscience Communication manuscript is to share insights from participatory research on water quality in the Rio Santa basin of Peru. More specifically, the focus is on (or should be on) the use of a novel methodology or tool – the Nuestro Rio photo elicitation app – to improve the participatory research process in several key ways. These topics are certainly within the scope of the journal, and I expect with minor to moderate revisions the manuscript will be suitable for publication as a GC Insights article.

**Specific Comments:**

My chief, and really only, concern with the manuscript at present is that the novel contribution – the Nuestro Rio app – is not always front and center. On their own, the four key lessons learned are not especially surprising. The value added, and the reason the manuscript merits publication as a GC Insight article, is due to the ability of the Nuestro Rio app to help address these four issues and improve research outcomes to the benefit of the research team and local communities in the basin. Hence, I would suggest the "lessons learned" be explicitly framed around how the app helps to address these issues/challenges to participatory water quality research, or can be improved upon in the future to do so. While this suggestion is one that cuts across the entire manuscript (i.e., it requires revising everything from the abstract through to the recommendations and conclusions), it is largely a matter of reframing the existing text, rather than wholesale rewriting.

*We thank you for reviewing this article, and for providing such positive and constructive comments. We really appreciated the feedback of changing the framing to bring the app into these lessons more to show how research like this can both be valuable while identifying challenges for future researchers to consider. We have done this for our revised version. This complete reframing has allowed us to show the focus of the manuscript on the app and lessons involved in developing and rolling it out, rather than the findings themselves (which can be explored in Rangecroft et al., 2023). Small changes have been made throughout the manuscript to achieve this reframing. We have reframed our abstract to also represent this focus. The abstract now reads as (lines 17 – 21):*

*"Here we share four key lessons from an interdisciplinary project (Nuestro Rio) that gathered community perspectives on local water quality in the Rio Santa basin (Peru) utilising a digital technological approach where we collected data via a novel photo elicitation app, supported with a field work campaign. The lessons explored in this article provide insights into challenges and opportunities for researchers considering developing technological tools for encouraging participation and engagement in marginalised communities."*

**Technical Corrections:**

These are all quite minor suggestions, so take them or leave them as you see fit!

In the first sentence of the abstract, you use the term "water security", while the research is centered on "water quality". You note the important water security implications in the introduction, but given the overarching focus of the paper is on water quality you might want to use this word in that first sentence.

*Thank you for this suggestion, we have removed the term water security from our abstract and in the introduction in our revised version.*

The acronym TEK is defined for "traditional ecological knowledge", but since you only use the acronym one more time thereafter, there's really no need to use the acronym in the first place.

*We have removed the acronym from our revised version.*

Make sure to define any concepts that you introduce (e.g., food-water-energy nexus), or alternatively avoid using them entirely if they are not germane to the focus of the manuscript.

*This is also a valuable point, thank you - we have removed this term completely.*

While the research project is certainly interdisciplinary, which you note several times, it is also fully transdisciplinary. You might want to include this term in conjunction with interdisciplinary, just to reinforce the strong connections between the research project and non-academic stakeholders.

*Thank you for making this point. We agree that our research is both inter- and transdisciplinary in nature, so we have now also added this in (for example, lines 84-85).*

I am familiar with the use of the term cosmovisión in Spanish, but I am not sure how widely used it is in English. Unless your aim is to communicate something deeper – drawing on deeper

cultural meaning embedded in the term – then I would replace it with "worldview" (or alternatively elaborate on the meaning of cosmovisión).

*Thank you for this suggestion - we have now replaced cosmovision with the term "worldview" in order to best communicate to an international readership (line 112).*

**References included in this document:**

*Rangecroft, S., Dextre, R.M., Richter, I., Grados Bueno, C.V., Kelly, C., Turin, C., Fuentealba, B., Camacho Hernandez, M., Morera, S., Martin, J., Guy, A. & Clason, C., (2023), Unravelling and understanding local perceptions of water quality in the Santa basin, Peru, Journal of Hydrology, 625(A), 129949, https://doi.org/10.1016/j.jhydrol.2023.129949*

*Saleem, U., Torabi Haghighi, A., Klöve, B., and Oussalah, M.: Citizen Science Applications for Water Quality Monitoring. A Review, EGUsphere [preprint], https://doi.org/10.5194/egusphere-2024-1170, 2024.*